# Nocturnal Transpiration May Be Associated with Foliar Nutrient Uptake

**DOI:** 10.3390/plants12030531

**Published:** 2023-01-24

**Authors:** Clara Vega, Chia-Ju Ellen Chi, Victoria Fernández, Juergen Burkhardt

**Affiliations:** 1Departamento de Sistemas y Recursos Naturales, Universidad Politécnica de Madrid, Ciudad Universitaria s/n, 28040 Madrid, Spain; 2Plant Nutrition Group, Institute of Crop Science and Resource Conservation, University of Bonn, Karlrobert-Kreiten-Strasse 13, D-53115 Bonn, Germany

**Keywords:** aerosols, nocturnal transpiration, nutrients, poplar, leaf anatomy

## Abstract

Aerosols can contribute to plant nutrition via foliar uptake. The conditions for this are best at night because the humidity is high and hygroscopic, saline deposits can deliquesce as a result. Still, stomata tend to be closed at night to avoid unproductive water loss. However, if needed, nutrients are on the leaf surface, and plants could benefit from nocturnal stomatal opening because it further increases humidity in the leaf boundary layer and allows for stomatal nutrient uptake. We tested this hypothesis on P-deficient soil by comparing the influence of ambient aerosols and additional foliar P application on nocturnal transpiration. We measured various related leaf parameters, such as the foliar water loss, minimum leaf conductance (g_min_), turgor loss point, carbon isotope ratio, contact angle, specific leaf area (SLA), tissue element concentration, and stomatal and cuticular characteristics. For untreated leaves grown in filtered, aerosol-free air (FA), nocturnal transpiration consistently decreased overnight, which was not observed for leaves grown in unfiltered ambient air (AA). Foliar application of a soluble P salt increased nocturnal transpiration for AA and FA leaves. Crusts on stomatal rims were shown by scanning electron microscopy, supporting the idea of stomatal uptake of deliquescent salts. Turgor loss point and leaf moisture content indicated a higher accumulation of solutes, due to foliar uptake by AA plants than FA plants. The hypothesis that deliquescent leaf surface salts may play a role in triggering nocturnal transpiration was supported by the results. Still, further experiments are required to characterize this phenomenon better.

## 1. Introduction

Atmospheric aerosols are liquid, solid, or mixed suspensions of heterogeneous size and composition. Atmospheric particles range in size from around 0.001 to 100 μm in diameter, and their composition depends on the origin and mechanism of formation [1]. Aerosols can be of natural or anthropogenic origin. Natural aerosols are mainly composed of sea salt, mineral abrasions, volcanic dust, or compounds emitted by plants, such as terpenoids [2]. Anthropogenic aerosols are composed of industrial dust, soot, and soil abrasion from agriculture and the construction industries or formed by gas-to-particle conversion from trace gases like ammonia and sulfur dioxide [3]. Aerosols deposited on leaf surfaces can be beneficial if they carry nutrients, but aerosols may also have adverse effects on the leaves [4,5]. Aerosols and other hygroscopic material on leaf surfaces can absorb or condense water [6,7,8]. Between 25% and 50% of European aerosols are ionic and hygroscopic [9], meaning they can deliquesce and form concentrated salt solutions at humidities below saturation. Previously, the indirect effect of aerosols on the water cycle [10], radiation [11], nutrient cycles [12], and their direct effect on leaf surfaces [13,14] have been assessed.

One possible effect of aerosols on leaf surfaces is the hydraulic activation of stomata (HAS). If aerosols are deposited close to open stomata, the solution can connect with the internal hydraulic system, leading to water loss and potentially affecting water use efficiency (WUE) [2,15]. On the other hand, HAS may also contribute to stomatal nutrient uptake. However, the results regarding HAS are controversial [14,16], and more studies are needed to understand this process better.

During the night, water loss via stomata is reduced, but stomata often do not close completely, allowing for gas exchange with the environment [17,18,19]. Nocturnal stomatal conductance (g_sw_) has been observed in plants of different species and environments [20,21,22,23], and the results point to nocturnal water loss, implying some advantages for plants under certain circumstances [24]. Different hypotheses have been assessed, suggesting that nocturnal transpiration may facilitate carbon fixation [25], intervene in oxygen and nutrient transport [17,26,27], or even help in drought [28,29] and salinity stress adaptation [21]. Nocturnal transpiration relations with nutrient uptake from the roots have been studied on different nutrients, such as nitrogen (N) [30,31,32], phosphorus (P) [33,34], and other different macro- and micronutrients [35]. Under conditions of ample water supply, an increase in nocturnal transpiration for nutrient-deficient plants was generally indicated in these studies.

Here, we tested the hypothesis that nocturnal transpiration and foliar nutrient uptake are connected. According to this hypothesis, nutrients on the leaf surface trigger stomatal opening. This increases the relative humidity in the leaf boundary layer and improves the mobility of hygroscopic substances that can be taken up via the stomata. Water loss by transpiration in unsaturated atmospheric conditions is inevitably caused by open stomata. However, this disadvantage may be overcompensated by the benefits of the uptake of deficient nutrients [15]. Thus, aerosol deposition might help maintain plant functions, especially in ecosystems with stable natural aerosols [15,36], where the species depend on the aerosol input of certain nutrients, especially phosphorus [37,38].

The present study focused on detecting nocturnal transpiration by continuous overnight monitoring of gas exchange. Different amounts of nutrient-containing leaf surface material were provided by cultivating plants in environments with unfiltered, AA, and filtered almost aerosol-free air, FA. As a third treatment, leaves were treated with an additional P fertilizer solution. Although foliar nutrient uptake was not measured directly, we used different physiological parameters, chemical analyses, and microscopical observations to describe processes and changes connected to the foliar uptake of P. These parameters included minimum leaf conductance (g_min_) and turgor loss point (π_tlp_), as indicators of drought tolerance [39,40,41], and foliar carbon isotope discrimination (δ^13^C) as a long-term measure of water use efficiency [42,43]. Anatomical leaf variables, such as leaf thickness, specific leaf area (SLA), and stomata characteristics, can be affected by aerosols [44,45,46]. SLA has previously been related to air pollutants and aerosols, showing an increase in polluted environments [47,48], and is related to resource conservation [49]. Aerosol deposition may also affect stomatal characteristics [50]. 

Clones of a hybrid poplar species (*Populus maxim. x nigra*) were chosen to avoid genetic variability [51]. Poplars, which have a fast-growing capacity, have been used as bioindicators of atmospheric trace elements [52,53,54] and are known to have high nocturnal transpiration rates [19]. We used plants growing in P-poor soil and compared greenhouse plants under normal ambient conditions with those under filtered air, with leaves grown within an almost aerosol-free greenhouse. In addition, we applied the lacking nutrient (P) to assess whether there was an increase in nocturnal transpiration.

## 2. Results

### 2.1. Nocturnal Transpiration

Each night, the transpiration of one leaf was continuously measured between 10 pm and 6 am of the following day. When making an average of the four nights, in which untreated FA leaves were measured, the vapor pressure deficit differed considerably, such as the measured transpiration rates and the calculated g_sw_ in absolute flux units (Figure 1A). The g_sw_ values were normalized to the respective g_sw_ value at 11 pm to concentrate on the nocturnal behavior of stomata. This decreased the mean variation coefficient from 27% to 2% and resulted in a continuous g_sw_ decrease of about 20% throughout the night, which was remarkably consistent among all four measured plants (biological replicates) (Figure 1B). There was a slight increase at about 5 am, close to sunrise. The consistency between untreated FA plants becomes more evident in Figure 1C, with lower y-axis resolution, where the g_sw_ values are compared to plants after P treatment. A conductance increase after 3 am, which resulted in about 10% higher than the 11 pm value, was shown by the latter (Figure 1C). Even higher variability was observed for the untreated AA plants (Figure 1D). The large error bars for the six replicates indicate a broad spectrum of trends from decreasing to increasing nocturnal conductance, with no continuous trend of the mean value throughout the night (Figure 1D). Additional P treatment decreased the variability and caused a moderate g_sw_ increase in the second half of the night (Figure 1E). Two AA leaves were repeatedly measured, for the first time, directly after applying the P solution and for the second time three weeks after. Decreasing nocturnal conductance was shown in both leaves during the first measurement (Figure 1F, dotted lines). Nocturnal conductance was constant or elevated in the second measurement (Figure 1F, straight, thick lines). A temporal threshold for the establishment of nocturnal transpiration was suggested by this behavior.

### 2.2. Physiological Parameters

An array of physiological variables, in addition to nocturnal transpiration, are shown in Table 1, including water loss, g_min_, turgor loss point (π_tlp_), carbon isotope ratio (δ^13^C), and SLA. Leaf water loss parameters, beta and tseca, did not differ between environments, but the dry basis moisture content was higher for AA than for FA (Table 1). Furthermore, leaves in an environment with aerosols retained more water, resulting in a high wet basis moisture content. However, leaves from both environments lost water at statistically indistinguishable rates (Figure 2), resulting in similar minimum leaf conductance (g_min_) for AA and FA leaves, as this parameter was calculated from the slopes. It was shown by leaf turgor loss point (π_tlp_) results that leaves with a reduced presence of aerosols (FA) had a higher (i.e., less negative) turgor loss point, compared to leaves in ambient air (AA). Carbon isotope ratios and SLA values were not significantly different between the AA and FA groups (Table 1).

### 2.3. Leaf Surface-Related Parameters

Several leaf surface variables are shown in Table 2, including stomatal parameters (e.g., length, width, area, and density) and contact angles with water (θ_w_). Stomatal adaxial size was consistently larger for FA leaves than for AA leaves. There were significant differences between groups for stomatal length, width, and area, but differences in pore width only occurred on the adaxial side. Stomatal density was higher on the abaxial side than on the adaxial side, but did not differ between environments. Adaxial leaf contact angles were lower in the AA greenhouse than in the FA one, thus making AA leaves more wettable (Table 2). The average leaf thickness was 2.00 ± 0.02 mm and did not differ between treatments.

On the scanning electron microscopy (SEM) images, the rims of guard cells appeared glabrous on FA leaves (Figure 3A,B), whereas crusts covered them on AA leaves (Figure 3C,D). Apart from the guard cells, no crystalline structures of single aerosols were observed on the AA or FA leaves. Flat areas, partially associated with crust-like structures, were available on the AA and FA leaves. However, they were more amorphous and covered other surface structures on the AA leaves (Figure 3).

Regarding transmission electron microscopy (TEM) images, the adaxial and abaxial sides had a similar structure, regardless of the treatment. Still, the cuticle (C, understood as a lipidized part of the cell wall), cell wall (CW), and total cell wall width were higher on the adaxial side (Figure 4). Furthermore, pectin was present in the middle section of the TEM images, as identified in other *Populus* species [55].

### 2.4. Leaf Mineral Element Concentration 

As expected, the leaf P concentration was low, and the measured 0.10 g 100 g^−1^ P (dry weight; DW) is considered deficient for poplar [56,57] (Table 3; Appendix A). None of the macronutrients were significantly affected by the aerosol presence, while for iron (Fe), Zn, and Cu, the AA leaves had significantly higher concentrations than FA leaves. Since the leaves were not washed, the increased presence of such metals may be linked to leaf surface aerosol deposition.

Differences in P status between plants treated with ammonium phosphate and untreated plants were addressed using the spectrometric OJIP method [58,59]. The most substantial differences were found in the I step of the OJIP transient for the dark-adapted leaves, which is a sign of foliar nutrient uptake (Figure 5, [58,59]).

## 3. Discussion

The main goal of this study was to test whether the presence of hygroscopic leaf surface material can trigger nocturnal transpiration. Testing this hypothesis is challenging, given the low magnitude of nocturnal transpiration and the limited knowledge we have regarding factors such as the physicochemical and signaling properties of leaf surfaces, aerosol loading, and nanoscale chemical interactions between leaf surfaces and aerosols. Clones from the same species were used to minimize the impact of genetic variability. The combination of aerosol exclusion (the AA/FA approach) and the additional amendment of aerosols were implemented to induce different amounts of leaf surface deposited material, thus combining two previously successful approaches for testing the ecophysiological impact of aerosols [14,60].

The decreasing nocturnal conductance of the untreated FA leaves was consistently shown by the results (Figure 1B). In contrast, the presence of particles, either from aerosol deposition (AA) or associated with the application of diammonium phosphate solution (“P treatment”), caused higher variability between plants and led to a constant or increasing nocturnal conductance (Figure 1C). Repeated measurements following P treatment indicated that a specific time was needed before aerosol amendment caused effective changes in nocturnal conductance (Figure 1F). The P treatment increased the variability between FA plants and decreased the difference between AA plants, but in both cases, it increased nocturnal conductance (Figure 1C–E). Furthermore, AA and FA leaves differed in turgor loss point, leaf moisture content (Table 1), stomatal dimensions, and appearance on the SEM images (Table 2). These results could contribute to a mechanistic interpretation regarding the influence of the aerosol on nocturnal transpiration.

We will consider the performance of untreated FA leaves, which reflect an aerosol-free standard of a pristine leaf surface, neither influenced by air pollution nor nutrient sprays, to interpret the obtained results. The consistency and the small variance of the normalized g_sw_ values (Figure 1B,C) can be considered quality proof for this “baseline” treatment. The contrasting situation, i.e., a large amount of surface material by adding P solution to AA leaves (Figure 1E), led to a relatively consistent increase in mean values with medium variance. There was a similar response for FA leaves that were treated with P solution (blue line in Figure 1C), indicating that the added P solution was the dominating factor, not the underlying amount of aerosols. The untreated AA leaves were between the clean FA surface and the P-treated leaves with considerable surface material. The single individuals in this group showed extremely high variability in nocturnal conductance and scattered overnight behavior (Figure 1D). A possible explanation is that the requirement for stomatal foliar uptake is to establish a thin aqueous film on a hydrophobic cuticular surface, which has to happen for each stomatal pore. This happens more efficiently with large amounts of salts, as for the P treatments, compared to small amounts from aerosol deposition alone. Establishing thin films also requires time, as seen from the changing behavior of treated leaves over time (Figure 1F). 

Concerning the hypothesis, it seems likely that HAS and the primary triggering factor are closely connected. HAS is a process in which hygroscopic particles adsorb water vapor from the environment and then deliquesce, i.e., they dissolve in the sorbed water, which may come from ambient air humidity or stomatal transpiration. Highly concentrated solutions are formed from deliquescent salts or evaporating P solutions, which, even on hydrophobic surfaces, may dynamically expand, depending on the nature of dissolved ions and their position within the Hofmeister series [61]. The expansion on the leaf surface is driven by humidity fluctuations [60,62] and HAS (i.e., the dynamic expansion into the stomata by salt creep), and the subsequent connection with the water film coming from the roots is an individual process for each stoma and requires time. 

The comparison between FA and AA stomata in SEM images (Figure 3) gives an impression of how these phenomena may be influenced by aerosols: the apparent absence of (crystalline) particles, together with the differences between the glabrous FA guard cells and the crust-like structures on AA guard cells, make salt crusts on stomatal AA structures likely. Similar crusts have also been observed for aerosols or salt solutions applied to other species [63]. In videos recorded by environmental scanning electron microscopes from pine needles, it has been shown how such crusts become deliquescent and form thin layers of highly concentrated salt solution that can enter stomatal structures [13]. The observed lower stomatal dimensions on the adaxial sides of AA, compared to FA leaves (Table 2), could be due to an adaptation to control the “desiccant” effect of aerosols [15,64]. Their larger aerosol mass implies intense dehydration by large aerosol particles, which are more frequently deposited on the upper side of leaves [65]. A wettability increase, due to aerosols, was reported by previous studies [15], and in the present study, results support this hypothesis, as the leaves with aerosols were more wettable. Contact angles were lower on the upper side than on the low side, which may also be due to higher amounts of particles, but could be supported by differing leaf surface waxes between the adaxial and abaxial sides (Table 2). 

A relation between nocturnal stomatal opening and nutrient acquisition from the soil was reported in previous studies [66,67]. Nutrient uptake during the night can add up to 51% of the total nutrient uptake, especially in the early night-time [68]. In some cases, transpiration could be comparable to day-time transpiration [69]. Higher nocturnal transpiration could occur in nutrient-deficit soils [32,34], although other studies observed the opposite [30] or even no effect of nocturnal transpiration on nutrient uptake [35,70]. In this study, we found that well-watered plants growing in P-poor soil opened their stomata at night after the foliar P treatment. However, this resulted in a moderate increase in nocturnal transpiration. This could mean that the process needs a stomatal opening mainly to establish a high humidity zone around the stomatal pore where deliquescence could happen.

As we confirmed with the results of the FluorPen (Figure 5), there was some foliar P uptake derived from the foliar treatment. Furthermore, the HAS hypothesis is supported by the lower leaf turgor loss point of AA, compared to FA plants (Table 1), with poplars from the same clone and under equal environmental conditions, apart from the aerosol concentration. The leaf turgor loss point (correctly “leaf water potential at turgor loss,” π_tlp_) [71] indicates the capacity of a plant to maintain cell turgor pressure during dehydration [41]. At the same time, the results depend on the species and environment [72,73,74]. Plants with more negative π_tlp_ would maintain stomatal conductance under lower water availability [75,76] and, thus, become more resistant to drought. A relation between aerosols and turgor loss points may develop via HAS [15]. For example, leaf turgor potential had been increased by saline aerosols, particularly when surfactants were added [77], which would alleviate HAS. In the present study, we found a lower (i.e., more negative) turgor loss point for the leaves growing in the AA, compared to the FA environment with aerosols (AA), which might also be a response to higher (diurnal) water loss and the resulting osmoprotection [78]. This observation agreed with the higher foliar moisture content (Table 1). High concentrations of macronutrients were not detected, but Fe, Zn, and Cu concentrations were significantly higher in AA than in FA leaves and might indicate higher foliar nutrient uptake (Table 3). Aerosol composition in the same two greenhouses was previously studied [14], and it was shown in their results that the aerosol load is mainly composed of nitrate, sulfate, and chloride. At the same time, Na, K, Mg, and ammonium are present in lower amounts. Iron is one of the most common components of the crust of Earth, and it is possible to find it as part of natural atmospheric aerosols [79]. However, the presence of Fe aerosols could also be due to anthropogenic causes, such as a highway [80] or rails nearby [81]. In this study, both greenhouses were located a few meters from a highway. Therefore, a high traffic load is indicated by detecting a higher concentration of Fe and Zn [80]. 

Regarding minimum leaf conductance (g_min_), it has been shown in previous studies that aerosols may cause leaky stomata and increase g_min_ [13]. However, the effect on g_min_ might depend on the nature of the aerosol and the species analyzed [14,82,83]. Here, g_min_ did not differ significantly with aerosol load (Table 1). This means that epidermal leaf transpiration, once stomata are closed (via the cuticle and any leaky stomata), is similar for both environments, supporting the interpretation that the nocturnal g_sw_ effects were indeed coming from more open stomata.

## 4. Materials and Methods

### 4.1. Plant Material and Experimental Conditions

The study was conducted between July to September 2021 in two greenhouses located in Bonn, Germany. One greenhouse was ventilated with filtered air (FA; about 1% of aerosols left), while the other greenhouse was ventilated with ambient air (AA). Trials were performed on 45 clone poplar seedlings (*Populus maxim. x nigra*) planted in a P-poor soil, one half growing in AA (N = 22) and the other half growing in FA (N = 23). The soil was constantly humid to avoid drought stress, and the air temperature and relative humidity (RH) were measured using a probe.

Physiological variables, such as nocturnal transpiration, cuticular transpiration rate, turgor loss point, and carbon isotope ratios of leaves, were measured. Furthermore, anatomical variables were also included, such as the contact angle, topography, and inner structure of poplar leaves.

### 4.2. Nocturnal Transpiration

Night measurements included both stomatal and cuticular water loss mechanisms described by the term “conductance” (g) [84], and nocturnal conductance is described using “g_N_.” The plants used for the study were not under drought stress, and genetic diversity did not affect either, as we used clones for the experiment.

Nocturnal stomatal conductance (g_sw_) was measured in one plant per night per instrument (LI-6800 Portable Photosynthesis System; LI-COR Biosciences, Lincoln, NE, USA). The conductance was measured every minute from 11 pm until 6 am the next day to ensure a stable concentration of CO_2_ supply and to avoid light from the environment. Chamber environmental settings were flow rate 300 µmol s^−1^, CO_2_sample_ 400 µmol mol^−1^ (stable CO_2_ concentration to prevent interaction with nocturnal stomatal conductance) [85], H_2_O off, fan speed 8,000 rpm, light off, and temperature. Stomatal conductance values were standardized with the original value at 11 pm and plotted using R Studio (R version 4.0.3) [86].

Nocturnal transpiration was measured in leaves from both environments (N = 14; AA: N = 8, FA: N = 6). After that, to evaluate the effect of nutrient foliar application on nocturnal transpiration, some leaves (N = 10; AA: N = 6, FA: N = 4) were simultaneously sprayed with diammonium phosphate solution -(NH_4_)_2_HPO_4_- (also, diammonium hydrogen phosphate; 1 g/L, diluted in deionized water) on both sides of leaves. The deliquescence humidity of this compound is 82.5% at 30 °C, i.e., at this RH, the compound forms a saturated solution [87].

### 4.3. Water Loss and Minimum Leaf Conductance

Minimum leaf conductance (g_min_) refers to the minimum value a detached leaf can reach when the stomata are completely closed because of desiccation stress [40]. In this study, g_min_ was estimated using one leaf from each plant (N = 32; AA: N = 16, FA: N = 16). Fully expanded leaves were quickly taken to the laboratory, with the petioles submerged in water. The petiole was sealed with melted paraffin wax (melting point 44 °C) to prevent water loss. A picture of each leaf was taken to determine the leaf area afterward. Subsequently, the leaves were weighted (fresh weight; FW) and placed in a growth chamber in a controlled environment, hanging on a clothesline by the petiole to dry. Leaves were periodically weighed every hour, for 9 h on the first and second days and 4 h on the third day. Temperature and RH in the growth chamber were recorded using a data logger (Tinytag, Gemini Data, Loggers, Chickester, UK). At the end of the trial, leaves were individually placed in paper envelopes and dried at 60 °C for 2 d for DW calculation.

Cuticular weight transpiration *g_min_* was determined after the leaf weight curve decreased linearly with time, leading to constant water loss. g_min_ is calculated based on intervals using the points where the water loss plot is linear and based on the slope and taking temperature, relative humidity, and saturation vapor pressure (VPsat) values into account. Both values should be similar. After that, we adjusted water content-time data to an exponential decay model. Water content on each point was calculated with the Equation of wet basis moisture content, as follows:(1)$${\mathrm{Moisture}\;(\%)}_{\mathrm{wet}}=\frac{\mathrm{FW}-\mathrm{DW}\;}{\mathrm{FW}}\;\times \;100$$
ranging from 0 to 100. The curve of moisture loss in poplar leaves growing in AA was compared to FA. Furthermore, three variables were also studied [88]: coefficient beta (b), which represents the slope of the decay model, being lower when the water loss curve drops more quickly; tseca, which is the time (min) required to reach the asymptote in the decay model; and moisture content, expressed in dry basis, as follows:(2)$${\mathrm{Moisture}\;(\%)}_{\mathrm{dry}}=\frac{\mathrm{FW}-\mathrm{DW}\;}{\mathrm{DW}}\;\times \;100$$
ranging from 0 to substantial percentages, depending on the situation. We also evaluated if temperature or RH influenced the data.

### 4.4. Leaf Turgor Loss Point

The leaf water potential at turgor loss (π_tlp_) has traditionally been calculated using pressure-volume (p–v) curves, but using a vapor pressure osmometer has proven to be a more efficient and reliable method [71]. For both greenhouses (N = 21; AA: N = 10, FA: N = 11), one leaf from each plant was collected and immediately placed with the petiole underwater and moved to the lab to rehydrate overnight, covered with an opaque bag. Each leaf was placed in a zipper bag with high humidity the following day. All of them were placed in a bag and stored in the fridge. For measurements, only one leaf sample should be taken out each time to prevent evaporation.

One sample disc was taken from the middle area of the leaf using a 6 mm diameter cork-borer and avoiding secondary veins. Leaf discs were immediately folded in an aluminum foil square (3 × 3 cm^2^) and frozen in liquid nitrogen for 2 min to fracture the cells and mix the contents. After that, the leaf disc was punctured to facilitate equilibrium, and the disc was quickly sealed in the vapor pressure osmometer (VAPRO 5600, Wescor, Inc, Logan, UT, USA). The osmolality (mmol kg^−1^) value was obtained after the osmometer reached equilibrium (5–10 measurement cycles). Then, the osmotic potential (π_o_) was calculated by using Van’t Hoff Equation (3), as follows:π_o_ = −Co × R × T(3)
where Co is the solute concentration (mmol kg^−1^), R is the universal gas constant (m^3^ MPa K^−1^ mol^−1^), and T is the temperature (K) [89]. Due to the strong correlation (R^2^ = 0.91) between π_o_ and π_tlp_ [71], π_tlp_ was calculated using Equation (4), as follows:π_tlp_ = −0.2554 + 1.1243 × π_o_(4)

### 4.5. Carbon Isotope Ratios

The carbon isotope ratios (δ^13^C) were measured with an isotope ratio mass spectrometer (IRMS, C-N-S Analyzer, and MS-2020; SerCon Ltd., Crewe, UK). Two to three leaves of each poplar seedling were collected from both greenhouses to obtain enough dry material (N = 21; AA: N = 10, FA: N = 11). The harvested leaves were oven-dried at 60 °C for one week to reach the absolute DW and were subsequently ground to a fine powder. Ground samples were weighted with an electronic micro-balance (M2P, Sartorius Lab Instruments GmbH & Co. KG, Göttingen, Germany). Therefore, 1 ± 0.1 mg was loaded into tin capsules. Tin capsules were placed inside the spectrometer, where they were subjected to the oxidation and purification process. The resulting gas stream was conducted using gas chromatography to take CO_2_ to the mass spectrometer and isotope detector. From the ratio of signals, which were collected at the detector, the ^13^C value was calculated, and the carbon isotope ratios (δ^13^C) were then calculated by comparison to a standard [60,90].

### 4.6. Leaf Anatomy

Leaf anatomy was evaluated through leaf thickness, leaf area, DW, and SLA (N = 32; AA: N = 16, FA: N = 16). Leaf thickness was measured with a micrometer, and leaf area was determined by photographing the leaf horizontally from a fixed height and angle on a grid paper. The processing of the image was performed with ImageJ1.52a (National Institutes of Health, Bethesda, MD, USA). SLA was calculated as the ratio between leaf area and DW.

### 4.7. Cuticular and Stomatal Characteristics

#### 4.7.1. Contact Angle Measurements

Contact angles were determined by depositing drops of water (2 µL volume) onto the abaxial and adaxial surfaces of polar leaves, using a drop shape analyzer (DSA30, Krüss, Hamburg, Germany). Fully expanded fresh leaves were harvested from both greenhouses (N = 12; AA: N = 6, FA: N = 6). Contact angles were taken from the right and left sides of the droplet, and they were measured by taking side-on optical photographs. Temperature and RH were controlled during the measurement.

#### 4.7.2. Electron Microscopy

The topography of poplar leaves from the two environments was analyzed by SEM. For SEM observations, fresh poplar leaves (N = 5; AA: N = 3, FA: N = 2) were cut into 4 mm^2^ pieces and mounted on stubs. Before observation, samples were gold-sputtered and then examined with a LEO Model 1450 VP (Variable Pressure; LEO Electron Microscopy Ltd., Clifton Road, UK), available at the Nees-Institute for the Biodiversity of Plants, University of Bonn, Germany. The specimen was observed under vacuum, with an accelerating voltage of 15 kV and a working distance of 12 mm.

For TEM observations, fresh poplar leaves were cut into 4 mm^2^ pieces and fixed in 2.5% glutaraldehyde solution overnight at 4 °C. Subsequently, the samples were washed three times in cacodylate buffer (osmolality of 380–400 mOsmol/l; pH 7.1) at 2 °C and postfixed with 1.5% OsO_4_ in the same buffer. Afterward, samples were dehydrated through an ascending isopropanol series (50%, 70%, 80%, 90%, 96%, and 100% twice) and embedded in Epon 812 resin (Glycidether 100; Carl Roth GmbH, Karlsruhe, Germany) using propylene oxide as a bridging solvent (50%, 75%, and 100% for three times). Samples were left in pure resin overnight (kept at 25 °C). Pure resin sample embedding was conducted in blocks incubated at 70 °C for 3 d. Finally, semi-thin sections were cut, mounted on nickel grids, and post-stained with Reynolds lead citrate (EMS, Hatfield, PA, USA) for 5 min before observation with a JEOL JEM 1010 electron microscope (Tokyo, Japan) at 80 kV and equipped with a CCD megaview camera [91].

The images obtained were processed using ImageJ1.52a (National Institutes of Health, Bethesda, MD, USA) software to measure stomatal density in total abaxial and adaxial surfaces and stomatal length, width, and area. As measurements were taken in summer, a specific shrinking of the samples could be expected. Still, as environmental conditions were equal in both environments, in case shrinking occurs, it might slightly affect values, but not the differences among groups.

### 4.8. Leaf Tissue Mineral Analysis

The concentration of mineral elements, both macronutrients and micronutrients (N, P, S, Mg, K, Ca, g 100g^−1^; B, Fe, Mn, Zn, Cu, Mo, mg kg^−1^), in leaves of the different species and treatments were determined (N = 16; AA: N = 8, FA: N = 8). After being taken to the laboratory, leaves were carefully washed in an acidulated 0.1% detergent solution and scrubbed with the fingers. They were then rinsed with abundant tap and distilled water. Washed leaves were oven-dried at 60 °C for 2 d, weighed, and ground before determining mineral elements. Nitrogen was measured with an elemental analyzer (TruSpec, Leco Corporation, St. Joseph, MI, USA). At the same time, the remaining elements were determined by inductively coupled plasma (ICP Optima 3000, Perkin-Elmer, Norwalk, CT, USA), following the UNE-EN ISO/ IEC 17025 standards for calibration and testing laboratories (CEBAS CSIC Analysis Service, Murcia, Spain).

The foliar nutrient status was also assessed by analyzing the shape of the time-dependent chlorophyll *a* fluorescence OJIP transient. When a plant is under P deficiency, the photosynthetic capacity decreases, which is reflected in a straighter and less steep I step of the chlorophyll *a* fluorescence transient (OJIP) [58,59]. We used a hand-held fluorometer (FluorPen; model FP-110/D, Photon Systems Instruments, Brno, Czech Republic; flash wavelength 455 nm) to calculate the OJIP curve. First, the youngest fully expanded leaf was dark acclimated for 15 min placing the leaf clip of the instrument in the lamina mid-section. Then, plant P status was determined using the relative variable fluorescence at time *t* by double normalizing transients between the minimum (*F*_0_) and maximum (*F*_m_) fluorescence values, according to Equation (5) [59,92], as follows:*V* (*t*) = [fluorescence (*t*) − *F*_0_]/(*F*_m_ − *F*_0_)(5)

The I step of the OJIP curve of plants with lower P status is straighter and has a less defined shape than plants with higher foliar P concentration [59].

### 4.9. Data Analysis

Statistical analyses were conducted using R software [86]. Data analysis included normality and homoscedasticity tests for the traits of study: leaf physiology-related parameters, leaf surface parameters, and mineral concentration analysis. Normality was assessed using the Shapiro–Wilk test [93] and homoscedasticity using the Fligner–Killeen test, if data were not normally distributed or F-test if data were normally distributed [94]. If data were normal and homoscedastic, an ANOVA and a post-hoc Tukey’s HSD test were conducted. If data was homoscedastic, but not normally distributed, a non-parametric Kruskal–Wallis test [95] and a pairwise Wilcoxon test [96] were conducted to evaluate group differences. Otherwise, robust tests like Yuen’s test were used, also in case outliers were detected, to determine if the differences among groups were significant [97].

## 5. Conclusions

Per se, stomatal transpiration means water loss for the plant. However, this loss is typically associated with simultaneous beneficial effects, such as stomatal CO_2_ uptake, long-distance xylem transport of water, mineral nutrients, and leaf cooling. Nocturnal transpiration has been observed for many C3 and C4 plants that cannot fix CO_2_ at night, but they may benefit by nutrient uptake from the soil, as reported in earlier experiments. This study suggests that foliar nutrient uptake can be another benefit associated with nocturnal transpiration. Plants may be stimulated by the availability of water-soluble nutrients on the leaf surface to open stomata at night. However, more detailed experiments with different plant species and experimental settings are required to clarify such phenomena.

## Figures and Tables

**Figure 1 plants-12-00531-f001:**
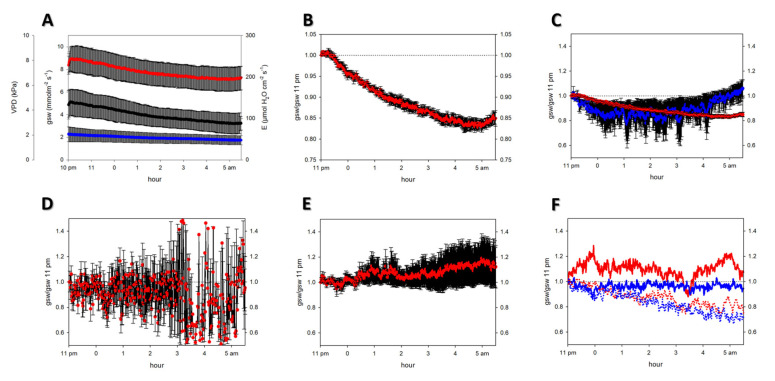
Nocturnal leaf conductance of poplar leaves. Error bars are standard errors. (**A**) Ambient vapor pressure deficit (blue, low line), measured transpiration (E) (black, middle line), and stomatal conductance for water vapor (g_sw_; red, upper line) of untreated leaves grown in filtered, aerosol-free air (FA). (**B**) g_sw_ of the same data set as in Figure 1A, now normalized to the g_sw_ value at 11 pm. (**C**–**E**) normalized g_sw_ of differently treated leaves with a y-axis range between 0.5 and 1.5. (**C**) g_sw_ of untreated (red) and P-treated (blue) FA leaves. The red line is identical to the red lines in Figure 1A,B, but now normalized and with a low y-axis resolution. (**D**) Untreated leaves grown under ambient aerosol conditions (AA). (**E**) AA leaves treated with P solution; (**F**) Repeated measurements of two AA leaves treated with P solution. Dotted lines: one day after treatment; solid lines: three weeks after treatment. For each analysis, the number of clones used was as follows: Figure 1A,B: N = 4; Figure 1C: N = 2 for the blue line and N = 4 for the red line; Figure 1D: N = 6; Figure 1E: N = 7; Figure 1F: N = 2.

**Figure 2 plants-12-00531-f002:**
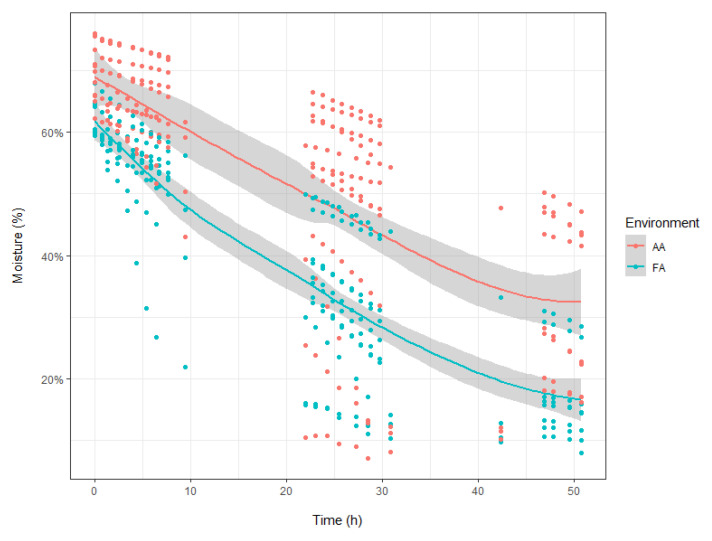
Wet basis moisture content (%) of poplar leaves during desiccation. Gray areas represent confidence intervals of 95%.

**Figure 3 plants-12-00531-f003:**
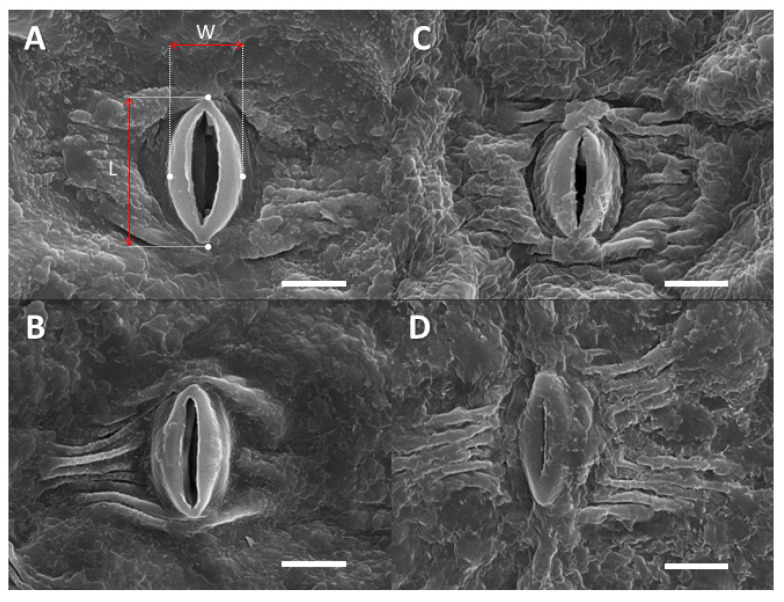
SEM micrographs of adaxial and abaxial stomata. Images correspond to (**A**) FA adaxial stomata, (**B**) FA abaxial stomata, (**C**) AA adaxial stomata, and (**D**) AA abaxial stomata. White bars correspond to 10 μm, and red arrows signal how stomatal width (W) and length (L) were measured. The bar size is 10 μm.

**Figure 4 plants-12-00531-f004:**
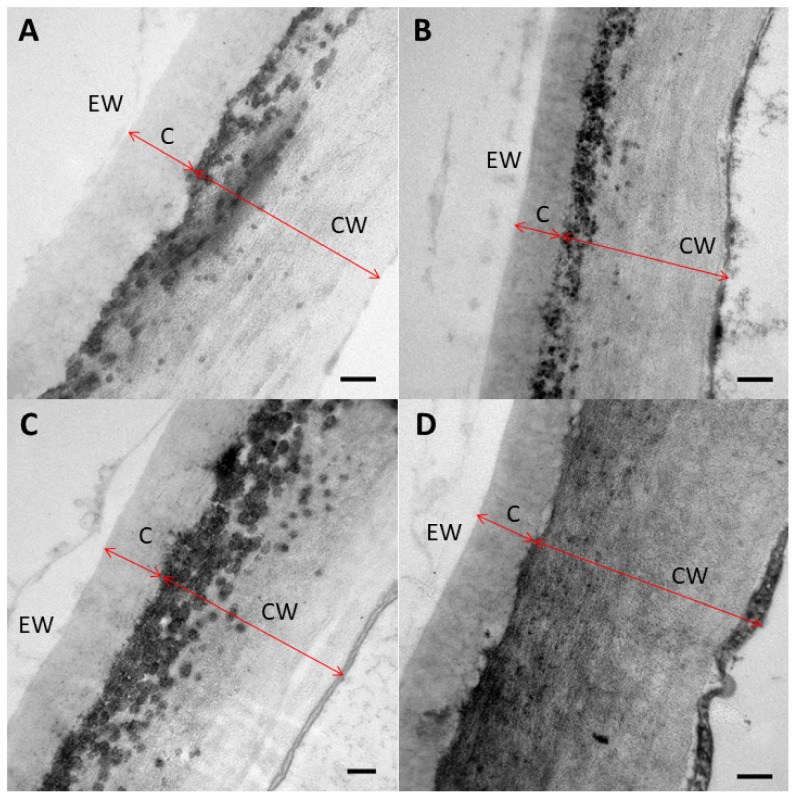
TEM micrographs of adaxial (**A**) and abaxial (**B**) side untreated FA leaves and adaxial (**C**) and abaxial (**D**) sides of ammonium-phosphate-treated FA leaves. Cuticle thickness (C), cell wall (CW) thickness, and epicuticular waxes (EW) are shown. The bar size is 0.2 µm.

**Figure 5 plants-12-00531-f005:**
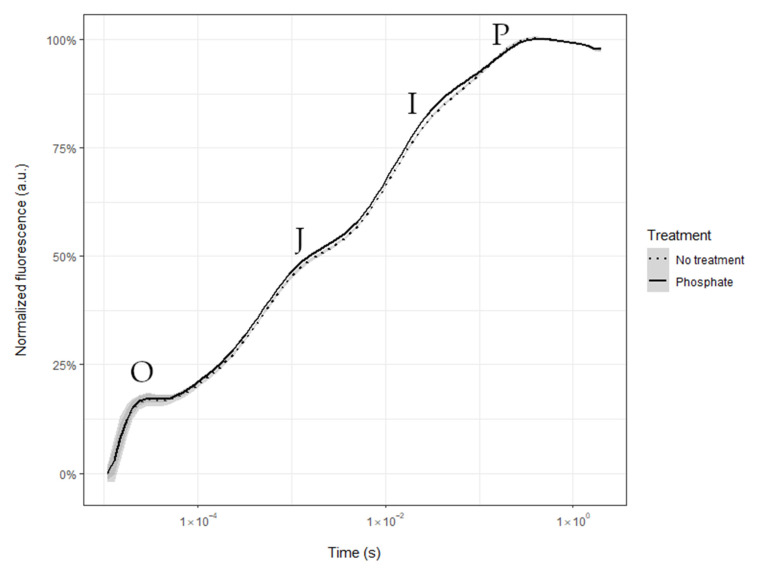
OJIP transients from treated (solid line) and untreated (dotted line) leaves. The I step for untreated plants has a straighter appearance in the OJIP transient than for treated plants, which is a sign of P deficiency. After normalization, the OJIP transients were measured in arbitrary units (see Materials and Methods).

**Table 1 plants-12-00531-t001:** Physiological parameters of poplar leaves for each environment. Beta and turgor loss point data were compared using ANOVA. Moisture (%) and SLA data were compared using the Kruskal–Wallis test, and the rest of the variables were evaluated using Yuen’s test. For each analysis, the number of clones used was N = 32 (g_min_), N = 16 (beta, tseca, moisture, and SLA), and N = 21 (turgor loss point and *δ*^13^C).

Environment	Beta (β)	tseca (h)	Moisture (%)_dry_	g_min_ (mmol m^−2^ s^−1^)	Turgor Loss Point π_tlp_ (MPa)	*δ*^13^C (‰)	SLA (m^2^/kg)
FA	−6.60 ± 0.59	29.3 ± 4.8	182.9 ± 1.6	2.54 ± 0.9	−2.50 ± 0.10	−29.2 ± 0.8	13.1 ± 0.3
AA	−7.18 ± 0.43	30.4 ± 4.3	206.3 ± 10.1	1.61 ± 0.3	−2.85 ± 0.07	−29.8 ± 0.6	14.4 ± 0.8
Differences			*		*		

Data are means ± SE. Significance code: 0.01 “*” 0.05.

**Table 2 plants-12-00531-t002:** Stomatal (St.) length, width, and area for FA and AA leaves, both adaxial and abaxial sides. Stomatal length and contact angle were compared using Yuen’s test, and stomatal width, area, and density data were compared using the Kruskal–Wallis test. For each analysis, the number of clones used was N = 60 (stomatal length, width, area, and density) and N = 90 (contact angle).

Environment	Leaf Side	St. Length (µm)	St. Width (µm)	St. Area (µm^2^)	Stomatal Density (stomata/mm^2^)	θ_w_ (°)
FA	Adaxial	23.9 ± 0.4 ^a^	9.97 ± 0.29 ^a^	187.8 ± 7.4 ^a^	34.8 ± 6.1 ^a^	94.4 ± 2.1
	Abaxial	19.7 ± 0.6 ^b^	7.92 ± 0.24 ^b^	123.0 ± 6.0 ^b^	246.7 ± 16.7 ^b^	103.21 ± 1.7
AA	Adaxial	17.2 ± 0.5 ^c^	8.92 ± 0.39 ^ab^	122.2 ± 8.1 ^b^	49.1 ± 6.9 ^a^	88.7 ± 2.0
	Abaxial	19.2 ± 0.3 ^b^	7.66 ± 0.31 ^b^	116.2 ± 6.1 ^b^	300.5 ± 28.5 ^b^	90.0 ± 2.7
	Environment	***		***		**
Differences	Leaf side	*	***	**	***	*
	Interaction between factors	***	***	***	**	

Data are means ± SE. Values marked with different letters were significant within columns (*p* ≤ 0.05). Significance codes: 0 “***” 0.001 “**” 0.01 “*” 0.05.

**Table 3 plants-12-00531-t003:** Concentration of macronutrients (g 100g^−1^ DW) and micronutrients (mg kg^−1^ DW) of poplar leaves growing in FA and AA. The concentration of P data was compared using the Kruskal–Wallis test, Zn concentration was compared using Yuen’s test, and the rest of the mineral concentration data were compared using ANOVA. For each analysis, the number of clones used was N = 8.

	Macronutrients (g 100g^−1^)						Micronutrients (mg kg^−1^)		
Environment	N	P	S	Mg	K	Ca	Fe	Zn	Cu
FA	1.44 ± 0.16	0.10 ± 0.01	0.20 ± 0.02	0.29 ± 0.03	1.00 ± 0.07	1.15 ± 0.12	41.3 ± 5.2	17.2 ± 1.1	3.6 ± 0.4
AA	1.70 ± 0.10	0.09 ± 0.01	0.24 ± 0.02	0.34 ± 0.03	1.07 ± 0.13	1.05 ± 0.15	74.9 ± 5.8	80.8 ± 15.3	4.2 ± 0.2
Differences							**	*	*

Data are means ± SE. Significance codes: 0.001 “**” 0.01 “*” 0.05.

## Data Availability

Not applicable.

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
