# Peer review of "Nocturnal Transpiration May Be Associated with Foliar Nutrient Uptake"

_plants, 2023, doi:10.3390/plants12030531_

Round 1
Reviewer 1 Report
The present manuscript aimed to reconcile nocturnal transpiration with foliar nutrient uptake. For instance, the present experiments described therein are very far from what could have been expected. There are no measurements of foliar nutrient uptake and many data for treated plants are lacking (for all the physiological parameters in fact !). The presentation of the results is highly questionable (normalization) and their interpretation are really far beyond the truth. The experiments are also lacking many controls (leaf temperature during gas exchange measurements and all the data for all the replicates for each conditions (FA, AA, FA treated, AA treated)). In fact, a kinetic experiment for the treatment was also expected. Is the gsw still higher after 2 nights after the treatment ? The result section is too short and do not explain in details the results, and their logical connection. But maybe this is because there is no connection at all. Please find below my comments.
Additional Major criticisms:
For all the figure legends, please add the number of replicates (biological I hope) and the statistical tests used. This should not be mentioned in the material and methods, but rather in each legend. So we can understand which data were treated with ANOVA or KRUSKAL.
In all the manuscript, the authors stated that nocturnal leaf conductance was similar to nocturnal transpiration. Unless the authors can show us the data for leaf gas exchanges (and not only the fig 1A for the control FA), please only refer to “conductance” in the entire manuscript.
Figure 1:
Stomatal aperture is dependent on the temperature and the VPD. Thus can you show us the data for leaf T°C during the experiment ? This criticism is for all the plants (FA, AA, treated and not treated). The data are only shown during the first night for treated plants (fig 1C) but the mean of 4 nights were used for the control (fig 1A). What happens during the others nights ? (2,3,4 for example). Regarding the differences in the figure C, I have serious doubt about the variations of the SE for the blue line, which seems to correspond to the fig 1B. Presumably, with such differences of variability, any statistical test would be irrelevant. Is this reason explaining why the authors did not provide any statistics for these data ? For the fig 1D and 1E, I found also very surprising that there are such differences of variability between the repetition.
The physiology related parameters were taken from which plants ? at which time regarding the night ? and which night was used ? And why the authors decided to measure such parameters.? What is the interest regarding the figure 1 ?
The table 1 only show data for AA and FA. What are the data for treated plants ?
Figure 2: I do not understand at all what is the interest of performing such experiment. Basically, you can compare initial FW and final DW without looking at the dessication process. Overall, it seems that AA plants were more irriguated than FA plants, thus uptaking more nutrients and using it to produce more biomass…But what is the point ? where are the data for treated plants ?
Figure 3 and table3: Where are the data for treated plants ? can you please put this data into bargraphs to help the reader to understand the statistical classification ? Besides this, FA plants have a higher stomatal aperture for the aDaxial face, which has 10-fold less stomata than the aBxial face. Thus, I do not think that it will change the respiration rate, regarding the variability in the measurements previously shown…
Table 3: The P leaf is low, but compare to what ?where are the data for treated plants, showing that the treatment resulted in an uptake correlating with changes in the respiration rate ?
L259: Here we go !!! You have no data on P uptake and you draw some conclusions….This is no acceptable in science. Perhaps, you could used labelled P (34P) and then quantify its assimilation after application and rinsing leaf surface using mass spectrometry ?
Additional Minor criticisms:
L292-302: what were the ranges for air temperature inside the chamber and for the humidity also ?
L307: Intriguing sentence. Do you have data for the relative water content ? or a soil volumetric water content probe to state such sentence ?
In the discussion, please refer to the important figures when you argue something. Because I cannot follow you, since many important information are lacking in the figures to draw possible conclusions.
L389: not the good title I think.
Reviewer 2 Report
Dear /
Editor-in-chief of Plants journal
Thank you for your invitation to review the review article titled “Nocturnal transpiration may be associated with foliar nutrient uptake”.
The article is interesting since it discuss scientific literature indicating some limitations of the traditional methods, and how existing studies support the need of a more ecologically sound approach to ERA based on the definition of the components of the receiving environment. The article is clear in the display of backgrounds as well as in results. Moreover, it also shows a relevant literature research even if the only 11 on 66 cited articles have been published in the last 5 years. A revision of language is also recommended. The article is suitable for publication
